# Transport of Nanoparticles into Plants and Their Detection Methods

**DOI:** 10.3390/nano14020131

**Published:** 2024-01-05

**Authors:** Anca Awal Sembada, I. Wuled Lenggoro

**Affiliations:** 1Department of Applied Physics and Chemical Engineering, Graduate School of Engineering, Tokyo University of Agriculture and Technology, Tokyo 184-8588, Japan; anca@st.go.tuat.ac.jp; 2School of Life Sciences and Technology, Bandung Institute of Technology, Bandung 40132, Indonesia

**Keywords:** assisted delivery, passive delivery, vascular bundles, localization, quantification, nanoparticle characterization, detection strategies

## Abstract

Nanoparticle transport into plants is an evolving field of research with diverse applications in agriculture and biotechnology. This article provides an overview of the challenges and prospects associated with the transport of nanoparticles in plants, focusing on delivery methods and the detection of nanoparticles within plant tissues. Passive and assisted delivery methods, including the use of roots and leaves as introduction sites, are discussed, along with their respective advantages and limitations. The barriers encountered in nanoparticle delivery to plants are highlighted, emphasizing the need for innovative approaches (e.g., the stem as a new recognition site) to optimize transport efficiency. In recent years, research efforts have intensified, leading to an evendeeper understanding of the intricate mechanisms governing the interaction of nanomaterials with plant tissues and cells. Investigations into the uptake pathways and translocation mechanisms within plants have revealed nuanced responses to different types of nanoparticles. Additionally, this article delves into the importance of detection methods for studying nanoparticle localization and quantification within plant tissues. Various techniques are presented as valuable tools for comprehensively understanding nanoparticle–plant interactions. The reliance on multiple detection methods for data validation is emphasized to enhance the reliability of the research findings. The future outlooks of this field are explored, including the potential use of alternative introduction sites, such as stems, and the continued development of nanoparticle formulations that improve adhesion and penetration. By addressing these challenges and fostering multidisciplinary research, the field of nanoparticle transport in plants is poised to make significant contributions to sustainable agriculture and environmental management.

## 1. Introduction

The transport of matter into plants, also known as plant uptake, is a crucial process for the growth and development of plants. Plants require various essential nutrients, water, and gases for their survival and growth. The primary processes facilitating the transport of matter into plants encompass water uptake and the assimilation of nutrients. Water is essential for plants as it serves as a solvent for nutrients and is involved in various physiological processes [1]. Water is absorbed by the roots of plants through a process called osmosis [2]. The root hairs, tiny hair-like structures on the roots, play a critical role in water uptake [3]. Additionally, Choi and Cho discovered that root hairs significantly enhance the soil retention capacity of seedling roots in *Arabidopsis thaliana* [3]. Plants rely on essential nutrients like macronutrients (e.g., nitrogen, phosphorus, and potassium) and micronutrients (e.g., iron, zinc, and copper) [4], which they absorb through their roots from the soil. Nutrient uptake is a complex process utilizing both passive and active transport mechanisms. Smaller ions typically enter root cells through passive diffusion driven by concentration gradients [5,6]. However, for nutrients found in lower soil concentrations, plants employ active transport, an energy-consuming process facilitated by specific root cell transport proteins. Examples of these proteins include K^+^ transporters and channels like *OsGORK*, *OsAKT1*, *OsHAK1*, *OsHAK5*, and *OsHAK21* [7]. Nutrients travel through two main pathways in plants. The symplastic pathway involves nutrient movement through the plant cells via plasmodesmata [8], while the apoplastic pathway involves movement through the cell walls and intercellular spaces [9]. After the roots absorb nutrients, the plant’s vascular system, composed of xylem and phloem, transports them to various tissues. Xylem carries water and nutrients from the roots to aerial parts, while phloem transports sugars and organic compounds throughout the plant [10]. In *Zea mays*, Wang et al. demonstrated varied localization and translocation patterns within the xylem and phloem [11]. Transpiration, the loss of water vapor through the stomata [12], creates tension in the plant’s vascular system, facilitating the upward pull of water and nutrients from the roots, known as transpiration pull [13]. Additionally, plants absorb carbon dioxide (CO_2_) through the stomata, mainly in the leaves, for photosynthesis, converting light energy into chemical energy and producing oxygen (O_2_) as a byproduct [14].

The transport in plants extends beyond water, nutrients, and gases, encompassing the movement of various particles or nanoparticles, as well as other exogenous materials. This includes genetic material such as double-stranded RNA (dsRNA). Nanoparticles can be transported into plants for several reasons, often as a result of environmental exposure or as a part of research and development efforts. The transport of nanoparticles into plants can have various implications, both beneficial and potentially concerning, depending on the type of nanoparticles and the intended purpose. Nanoparticles can be classified based on both their chemistry and size [15]. The chemical composition of nanoparticles dictates their inherent properties, such as optical, electrical, magnetic, and catalytic characteristics [16,17]. Different materials exhibit unique behaviors and functions. Additionally, particle size also directly influences their physical and chemical properties due to quantum effects and an increased surface-to-volume ratio [18]. In many cases, a combination of both chemistry and size is considered to tailor nanoparticles for specific applications. Nanoparticles have dual applications in delivering nutrients, fertilizers, and genetic materials to plants, offering significant benefits in agriculture and biotechnology [19,20]. Designed for controlled nutrient release, nanoparticles, like the nanoU-NPK fertilizer (Ca 23.3%, P 10.1%, K 1.5%, NO_3_ 2.9%, urea 4.8%), minimize over-fertilization and enhance nutrient use efficiency [21]. This controlled release also mitigates environmental impacts related to nutrient leaching and runoff. Additionally, nanoparticles act as carriers for genetic materials, facilitating genetic modification, gene silencing, or gene editing [20,22]. They deliver transgenes or genetic constructs into plant cells for trait enhancement or modification [22]. Small RNA molecules, like small interfering RNA (siRNA), delivered by nanoparticles, initiate RNA interference (RNAi) for targeted gene regulation or virus resistance [23]. Combining nutrient and genetic material delivery through nanoparticles is particularly valuable in modern agriculture, addressing challenges and contributing to crop improvement.

In this review, we will provide an overview of the transport of nanoparticles in plants and the methods employed for their detection. Additionally, we will outline the challenges associated with current methodologies and explore potential future directions.

## 2. Transport of Nanoparticles into Plants

Understanding the transport of nanoparticles into plants is crucial for harnessing their potential in agriculture and biotechnology. The journey of nanoparticles from external environments to internal plant tissues involves intricate processes influenced by both the properties of the nanoparticles and the characteristics of plant structures. Nanoparticles can be designed to transport specific materials, compounds, or genetic information within plants, acting as carriers to deliver these payloads to target locations. The controlled delivery of nanoparticles can be advantageous for precise and efficient plant applications. Nanoparticles can be used for fertilizer or nutrient delivery, genetic material delivery, and pesticide or herbicide delivery (Figure 1). Nanoparticles can be used to transport essential nutrients, including micronutrients such as iron, zinc, and other trace elements, into plant cells [21,24,25]. Tombuloglu et al. successfully synthesized composites of micronutrient nanoparticles (NPs) and applied them to *Hordeum vulgare* L. [25]. They demonstrated the effective incorporation of these micronutrients into plant tissue. The transportation of these NPs significantly increased the quantity of elements in both the root and leaf tissues. Specifically, the contents of Fe, Zn, and Cu were raised to approximately 5, 3, and 18 times higher than the control, respectively [25]. This approach is known as nanoparticle-mediated nutrient delivery and is aimed at enhancing the nutrient uptake and utilization of plants. Nanoparticles, due to their small size, can be taken up by plant roots or leaves. The nanoparticles may enter plant cells through various mechanisms, such as diffusion, endocytosis, or direct uptake through ion channels and transporters [7,26]. Once inside the plant cells, the nanoparticles release the encapsulated nutrients. This controlled release ensures that the nutrients are made available within the plant at a rate that matches the plant’s needs [25]. By improving nutrient uptake and utilization, nanoparticle-mediated nutrient delivery can reduce the number of traditional fertilizers needed, minimizing the environmental impact associated with excess nutrient runoff and leaching.

Nanoparticles can also serve as vehicles for delivering genetic material, such as DNA, RNA, or small interfering RNA (siRNA), into plant cells [22,23]. This technology, often referred to as “nanoparticle-mediated gene delivery”, has several applications in plant biotechnology and agriculture. Nanoparticles are engineered to encapsulate, bind, or complex with genetic material. These nanoparticles are designed to protect the genetic material from degradation and facilitate its delivery into plant cells [27]. Wang et al. used several types of NPs (CS, PEI, protamine, CQD, PAMAM, and CSC) to deliver dsRNA against rice sheath blight (*Rhizoctonia solani*) in *Oryza sativa* L. [28]. These nanoparticles could protect dsRNA from degradation by nucleases. Nanoparticles are introduced to plant tissues through various methods, such as direct injection into plant cells, root soaking, or foliar spray. Once inside the plant cells, the nanoparticles release the encapsulated genetic material. This can be achieved through controlled release mechanisms or by breaking down the nanoparticle complexes under specific conditions. Nanoparticles can be used to transport pesticides or herbicides to specific target areas within plants [29]. This approach can enhance the precision and efficiency of pesticide application and reduce the environmental impact associated with conventional spray applications. Nanoparticles can be designed to target specific plant tissues or cell types, ensuring that the pesticides or herbicides are delivered directly to the intended areas, such as the leaves, stem, or root system [29,30]. A water-soluble chitosan (CS) derivative (N-(2-hydroxyl)propyl-3-trimethyl ammonium CS chloride, HTCC) was successfully capped on the surface of pyraclostrobin-loaded MSNs by Cao et al. [31]. This particle directly targeted *Phomopsis asparagi* (Sacc.) and had fungicidal activity against it. By improving the targeted delivery of pesticides and herbicides, nanoparticle-mediated delivery can reduce the amount of chemicals needed, minimize off-target effects, and reduce environmental pollution and contamination.

Nanoparticles have also shown potential impacts on plant development and growth, and their applications in agriculture are an active area of research. Nanoparticles can influence seed germination rates and early plant growth [32,33]. They may enhance seedling vigor and promote healthier plant establishment. Our previous study showed that the use of SiO_2_ NPs could improve tomato (*Solanum lycopersicum* var. Momotaro) seed germination [33]. Some nanoparticles have been studied for their potential to enhance plant stress tolerance, such as resistance to drought, salinity, or heavy metal toxicity [34]. These nanoparticles may act as stress mitigators and improve overall plant health. The presence of nanoparticles in soil can also influence microbial communities and soil health [35]. The impact depends on the type of nanoparticles used and their interactions with soil microorganisms. Some nanoparticles may exhibit phytotoxic effects. This can result in stunted growth, reduced photosynthesis, or other negative impacts on plant health [36]. The fate of nanoparticles in the environment, including their persistence and potential for leaching into water sources, is an important consideration for sustainable agricultural practices. Balancing the potential benefits with environmental and human health considerations is crucial for the sustainable integration of nanotechnology in agriculture.

### 2.1. Types of Nanoparticles Used for Transport

Various types of nanoparticles are used for transport in plants, depending on the specific application and the payload (substance being transported). Different nanoparticles possess unique properties that make them suitable for diverse purposes in plant biology and agriculture. The choice of nanoparticle type in plant-related applications depends on several critical factors, as listed in Table 1. The careful consideration of these factors is essential when choosing the appropriate nanoparticle type for specific plant-related applications. The goal is to ensure that the selected nanoparticles align with the objectives of the application, provide benefits, and minimize potential risks to plants and the environment.

Nanoparticles can be categorized into commercial (engineered) and biogenic (naturally occurring) based on their origin [37]. Commercial nanoparticles, also referred to as engineered nanoparticles, are particles that are intentionally designed, synthesized, and produced with specific characteristics for various industrial, technological, agricultural, or consumer applications such as metal nanoparticles, metal oxide nanoparticles, carbon nanotubes, quantum dots, nanocomposites, and polymeric nanoparticles [37,38]. These nanoparticles are created through engineering processes to achieve desired properties at the nanoscale, typically ranging from 1 to 100 nanometers. The intentional manipulation of the size, shape, surface properties, and composition distinguishes engineered nanoparticles from naturally occurring nanoparticles. On the other hand, biogenic nanoparticles refer to nanoparticles that are naturally formed or synthesized by living organisms or natural processes without human intervention. Unlike engineered nanoparticles, which are intentionally produced by humans for specific purposes, biogenic nanoparticles occur naturally because of biological and environmental processes [39]. These nanoparticles can be found in various natural sources and are often associated with specific organisms or geological phenomena such as bacterial nanoparticles, fungal nanoparticles, volcanic ash, clay minerals, and wildfire ash [39].

**Table 1 nanomaterials-14-00131-t001:** Factors influencing nanoparticle selection for plant transport-related applications.

Factor	Description
Desired application	Different applications, such as nutrient delivery, genetic material transfer, or pesticide transport, require specific nanoparticle types with suitable properties [25,28,31].
Payload type	The nature of the payload, whether it is nutrients, genetic material (DNA or RNA), pesticides, or other substances, influences the selection of the appropriate nanoparticle [23,24,29].
Payload size and solubility	The size and solubility of the payload may determine the choice of nanoparticle, as some nanoparticles are better suited for carrying particular types of cargo [40].
Targeted delivery	If precise delivery to specific plant tissues or cells is required, the nanoparticle type should allow for targeted delivery [41].
Biocompatibility	Some applications, such as those involving genetic material delivery or interactions with living organisms, necessitate biocompatible nanoparticles [42].
Environmental considerations	The environmental impact of nanoparticle use, including factors like biodegradability and safety, is crucial in agriculture and ecological applications [43]. In addition to these considerations, environmental conditions such as soil pH, temperature, and relative humidity play pivotal roles in determining the fate and impact of nanoparticles [44].
Size and shape	The size and shape of nanoparticles can influence their ability to enter plant cells or tissues. In some cases, specific shapes or sizes may be more effective [29,45].
Crop or plant type	Different plant species or crops may have varying requirements and responses to nanoparticle-based applications, affecting the selection of nanoparticle types [46].

There are numerous types of nanoparticles, and the field of nanotechnology continues to evolve, leading to the development of new types and functionalities. Examples of nanoparticles used for transport in plants are presented in Table 2 along with their introduction site. Metals and metal oxides are types of nanoparticles that are commonly used in various plant transportation applications.

Research on the interaction of nanomaterials with plants has mainly focused on phytotoxicology. At the same time, less research has been conducted on positive effects such as increasing crop productivity and enhancing plant resistance, and research on beneficial effects on plants is still incomplete. The entry of nanoparticles into plants, especially at high concentrations, can lead to phytotoxicity. Geisler-Lee et al. showed that Ag NPs could exert detrimental effects on *A. thaliana*, but the phytotoxic effect of Ag NPs could not be fully explained by the released silver ions [50]. Another study also indicated that the internalization and upward translocation of ZnO NPs by *Lolium perenne* could significantly reduce the biomass and cause damage to root epidermal and cortical cells [54]. Some nanoparticles, such as certain metal and metal oxide nanoparticles, can generate reactive oxygen species (ROS) when they enter plant cells [83]. Elevated ROS levels can damage cell structures, disrupt cellular functions, and cause oxidative stress, leading to plant injury or even cell death [83,84]. High nanoparticle concentrations can disrupt the integrity of plant cell membranes [85]. This can lead to the leakage of cellular contents and negatively impact cell viability. Rossi et al. stated that NPs could also influence plant symplastic pathways by altering ion transport activity or root cell membrane integrity [86]. Excessive nanoparticle accumulation can interfere with the uptake and distribution of essential nutrients, disrupting nutrient balance in plants and causing nutrient deficiencies or toxicities. High nanoparticle concentrations may also physically obstruct the plant’s nutrient and water transport systems, impeding the movement of substances throughout the plant [26]. Plants exposed to high nanoparticle concentrations may activate stress responses, diverting resources away from growth and development and reducing overall plant health [46,87]. The phytotoxicity of nanoparticles can vary depending on factors such as nanoparticle type, size, shape, surface properties, and the plant species involved. Therefore, selecting the appropriate concentration of nanoparticles is a critical decision, as it can significantly impact the performance and effectiveness of nanoparticle transport.

Nanoparticles, whether naturally occurring or engineered, can have various negative effects on plants. The impact of nanoparticles on plants depends on factors such as the type of nanoparticles, their concentration, exposure duration, and the specific plant species [88]. Nanoparticles can exhibit phytotoxic effects, leading to damage being caused to the plant cells, tissues, and overall plant structure. This can result in stunted growth, reduced biomass, and compromised plant health [36]. Exposure to certain nanoparticles can alter the root morphology and function. This includes changes in the root length, surface area, and the structure of root hairs, which can impact nutrient and water uptake [89]. Nanoparticles can induce oxidative stress in plants by generating reactive oxygen species (ROS). Elevated ROS levels can lead to the damage of cellular components, such as membranes, proteins, and DNA, affecting plant health [90]. Some nanoparticles may persist in the environment, leading to long-term exposure for plants. Additionally, there is a concern about the potential bioaccumulation of nanoparticles in plant tissues, which can have implications for organisms higher up the food chain [91].

### 2.2. Modes of Transport of Nanoparticles into Plants

Nanoparticle transport in plants can be categorized based on two main mechanisms, i.e., assisted delivery and passive delivery. When referring to assisted delivery in the context of transporting nanoparticles to the plant body, it means that external power or forces are applied to facilitate the transport of nanoparticles into the plant tissues [20]. This external power or force assists in the delivery process, often overcoming barriers that would hinder passive transport [92]. Examples of assisted delivery techniques in plant nanoparticle applications are shown in Figure 2 and described briefly in Table 3. The biolistic (gene gun) is a device that uses an external force, such as compressed gas or helium, to propel nanoparticles [93,94]. Rajkumari et al. reported the use of Ag NPs as gene carriers, replacing Au microcarriers for biolistic gene delivery in *Nicotiana tabacum* L., and showed that the transformation efficiency was significantly higher with Ag NPs than Au microparticles as carriers [94]. Sonoporation involves the application of ultrasound waves to create temporary pores in plant cell membranes, allowing nanoparticles to enter the cells [95]. The external force of ultrasound assists in the delivery process. Zolghadrnasab et al. showed that ultrasonic treatment provides an economical and straightforward approach for poly-ethyleneimine (PEI)-coated mesoporous silica nanoparticles (MSNs) transferring into plant cells without any need for complicated devices and without concerns about safety issues [96]. Magnetic nanoparticles can be guided to specific plant tissues using external magnetic fields, effectively assisting in the targeted delivery of these nanoparticles [97]. Characterizing the intrinsic magnetic properties of nanoparticles involves understanding their behavior at the bulk level as well as at the level of individual molecules. Various techniques are employed for both bulk and single-molecule investigations. Bulk techniques include vibrating sample magnetometry (VSM), superconducting quantum interference device (SQUID), magnetic resonance imaging (MRI), Mössbauer spectroscopy, and X-ray magnetic circular dichroism (XMCD) [98,99]. Single-molecule techniques include magnetic force microscopy (MFM), scanning tunneling microscopy (STM), electron magnetic circular dichroism (EMCD), single-particle magnetometry, and fluorescence-based techniques [100,101]. Electroporation involves applying external electric fields to plant cells, creating temporary pores in the cell membranes [102]. This method uses an external electrical power source to facilitate nanoparticle entry. Microinjection, combined with external manipulation, can be used to introduce nanoparticles into specific plant cells [103]. Most of the assisted delivery methods are used for transporting nanoparticles in vitro (outside living organisms, typically in a controlled laboratory setting) as indicated in Figure 2. Nanoparticles can be introduced to plant embryos cultivated in a controlled laboratory setting [104]. Plant embryos, which are the earliest stages of plant development, offer a convenient point of entry for introducing nanoparticles that can influence the growth and characteristics of the resulting plant. Nanoparticles can also be introduced to plant callus cultures, which are undifferentiated masses of plant cells grown in vitro [105]. Plant callus cultures are typically initiated from explants, such as leaf pieces, stem segments, or immature embryos. These explants are sterilized and placed on a suitable culture medium containing plant growth regulators to induce callus formation. Introducing nanoparticles to plant protoplasts is also a technique that is commonly used in plant biotechnology. Protoplasts are plant cells that have had their cell walls enzymatically removed, leaving behind the cell membrane and cytoplasm [106].

Assisted delivery methods can facilitate the delivery of relatively larger particles due to the external forces involved, as indicated in Table 3. These methods are valuable for overcoming natural barriers and transporting particles efficiently. However, there are still many big issues related to this transport method, especially regarding their scalability to larger scales. Therefore, passive delivery still has advantages over it. In a controlled laboratory setting, it is easier to maintain uniform conditions for assisted delivery, ensuring consistency in nanoparticle uptake [107]. On larger scales, achieving uniformity across a field or agricultural area becomes more challenging. Variability in environmental conditions, soil composition, and plant physiology can affect the effectiveness of delivery methods [108]. Assisted delivery in vitro also allows for precise control over factors such as temperature, humidity, and nutrient levels. However, real-world conditions vary widely in agricultural settings. Adapting in vitro strategies to diverse environments becomes complex, and factors like wind, rainfall, and temperature fluctuations can impact the efficacy of nanoparticle delivery [109]. Implementing these assisted delivery methods on a large scale also requires significant resources and may not be economically feasible. Factors such as the cost of materials, equipment, and labor must be considered for practical application in agriculture. The large-scale implementation of assisted delivery may face regulatory challenges related to environmental impact, safety, and ethical considerations [110]. Demonstrating the safety and environmental compatibility of nanoparticle delivery systems becomes essential.

**Table 3 nanomaterials-14-00131-t003:** Examples of assisted delivery techniques for introducing nanoparticles into plants.

Method	NPs	Size (nm)	Plants	Target	Ref.
Biolistic (gene gun)	Ag	100	*Nicotiana tabacum*	Leaf explants	[94]
	Au	712 ± 95	*Nicotiana benthamiana*	Leaf explants	[106]
	Au-MSN	600	*Nicotiana tabacum*, Teosinte	Leaf explants	[111]
	Au-MSN	600	*Zea mays*	Embryos	[103]
	Fe	255 ± 170	*Nicotiana benthamiana*	Leaf explants	[106]
Sonoporation	hPAMAM-G2	123 ± 21	*Medicago sativa*	Cells	[112]
	PEI-MSN	100 ± 87	*Nicotiana tabacum*	Cells	[96]
Magnetic field	γ-Fe_2_O_3_	21.2 ± 3	*Zea mays*	Roots	[113]
	Carbon-coated iron	10–50	*Cucurbita pepo*	Roots	[114]
	Fe_2_O_3_	10	*Solanum lycopersicum*	Roots	[115]
	Fe_3_O_4_	13	*Hordeum vulgare*	Roots	[116]
Electroporation	CPNs	60–80	Tobacco	Protoplasts	[105]
Microinjection	mGNPs	20–30	*Brassica napus*	Cells	[117]
	SWNTs	500	*Nicotiana tabacum*	Cells	[118]

In contrast to assisted delivery, passive delivery in the context of nanoparticle transport in plants typically refers to the movement of nanoparticles through natural routes without the assistance of external forces or active mechanisms. It encompasses various methods through which nanoparticles enter the plant body, often via sites such as roots and leaves. In summary, assisted delivery and passive delivery are different concepts. Assisted delivery involves an intervention to enhance nanoparticle uptake, while passive delivery relies on natural processes, without external assistance. These approaches cater to different research goals and applications in the field of nanotechnology for plant science. These passive delivery routes include root uptake and foliar uptake [29]. Plants naturally absorb water and dissolved substances, and nanoparticles can enter the plant through the roots as they take up water and nutrients [26,29]. This is a common method for delivering nanoparticles to plants in vivo. When discussing in vivo delivery in the context of plants, it typically involves the intact, living plant. In vivo experiments aim to study the interaction between nanoparticles and the entire living organism, considering the complexities of the plant’s biological systems [119]. Nanoparticles can also be sprayed or applied to the leaves of plants [26,120]. Lian et al. showed that the response of *Zea mays* L. to Cd and TiO_2_ NPs was highly dependent on the exposure mode. They reported that leaf exposure provided more benefits than root exposure [120]. Some nanoparticles, like those used in foliar fertilizers or pesticides, can be taken up by the plant through the stomata (small openings on leaf surfaces) or through the leaf cuticle [29]. Examples of these two methods are shown in Table 1 above, which use the in vivo nanoparticle transport route (Figure 3). After nanoparticles enter the roots or leaves of a plant, they can be transported throughout the plant’s vascular system, which includes the xylem and phloem in the stem [121]. The vascular system plays a crucial role in the distribution of water, nutrients, and various substances, including nanoparticles, to different parts of the plant.

### 2.3. Challenges during Nanoparticle Transport

One of the primary challenges of using assisted delivery techniques is the potential for plant cell damage [122]. Assisted delivery methods, especially those involving physical forces or electrical pulses, can cause physical stress or damage to the target plant cells [123]. This may result in cell death, reduced viability, or altered cell function. Each assisted delivery method often requires the careful optimization of parameters. Finding the optimal parameters for different applications can be labor-intensive and may involve trial and error. Some assisted delivery methods may raise safety concerns due to potential unintended biological effects. Some assisted delivery methods may also be less suitable for in vivo applications due to challenges related to safety, depth of penetration, and the potential for systemic effects. A description of the advantages and disadvantages of each assisted delivery method can be seen in Table 4. Some of the disadvantages associated with assisted delivery methods are precisely what make passive delivery methods preferable in certain situations. Passive delivery methods are often favored when conducting studies in living plants, particularly when the goal is to minimize risks to plant health and create conditions that more closely resemble real-world scenarios.

Passive delivery methods in plants have several advantages over assisted delivery methods, especially in certain applications and contexts. Passive delivery methods do not require the application of external forces, such as mechanical or electrical forces, which can potentially damage plant cells or tissues [27,122]. This results in less invasive effects on the plant, reducing the risk of cell damage or stress. Characterizing the mechanical and electrical properties of nanoparticles is essential for understanding their behavior and performance, especially when they interact with plant tissues [135]. Mechanical properties at the bulk level are characterized by atomic force microscopy (AFM) and dynamic mechanical analysis (DMA) [136]. In contrast, AFM-based nanoindentation and scanning probe microscopy (SPM) are the techniques commonly used to characterize at the nanoscale level [137], and techniques used to characterize electrical properties at the bulk level are impedance spectroscopy and four-point probe measurement [138]. On the other hand, techniques used to characterize at the nanoscale level are conductive AFM (cAFM), Kelvin probe force microscopy (KPFM), and transmission electron microscopy (TEM) with energy-dispersive X-ray spectroscopy (EDS) [139]. Passive delivery methods are often simpler and more straightforward to implement. They do not require specialized equipment or complex optimization processes, making them accessible to a wider range of researchers and practitioners. Passive delivery methods are generally more cost-effective because they do not necessitate the use of expensive equipment or consumables. However, there are several challenges, particularly when it comes to particle size limitations. Passive delivery methods are often constrained by the physical characteristics of the nanoparticles being transported [140]. In the case of roots and leaves, the size of particles that can be transported is limited by the size of pores, channels, or structures within the plant tissue [141]. Large particles may encounter limitations in passing through these structures, thereby constraining the efficacy of passive transport. The examples presented in Table 1 indicate that the majority of the particle sizes utilized are below 40 nm. Particles with sizes above 40 nm may not be accommodated through passive delivery via roots and leaves. Plant cells have rigid cell walls, and the pore size diameter ranges from less than 10 nm in most pores to a rare maximum size of 40 nm [142,143,144]. McCann et al. observed that intact cell walls of *Allium cepa* var. Jumbo generally measure 10–20 nm [142]. The average cell wall size was 30 nm in width, which was visualized using spectroscopic methods [145]. The same is true for foliar uptake, where a waxy cuticle in leaves acts as a barrier for particles entering the plant [146]. Many particles, particularly larger ones, will not be able to pass through the stomata (microscopic openings) on the leaf surface, as these typically have diameters of less than 10 nm to 20 nm [147]. Passive delivery methods often involve incorporating nanoparticles into the growth medium, soil, or other components of the plant’s environment during specific developmental stages, such as seed germination or seedling growth, as indicated in Table 5. Passive delivery via root uptake and foliar uptake has its own set of advantages and disadvantages, as also listed in Table 5. Exploring alternative introduction sites for passive transport in plants is a worthwhile direction for future research and development. It allows for the customization of delivery methods and the potential to overcome limitations associated with roots and stems.

## 3. Detection of Nanoparticles in Plants

Selecting the appropriate detection methods for nanoparticles in plants is crucial for various applications, including understanding nanoparticle uptake, distribution, and potential impacts on plant health and the environment [125,157]. The choice of detection method should depend on the specific research objectives, the characteristics of the nanoparticles, and the resources available. Often, a combination of techniques is employed to gain a comprehensive understanding of nanoparticle transport in plants [158,159]. Deng et al. studied the uptake of titanium dioxide NPs in *Oryza sativa* L. tissues using multiple orthogonal techniques: electron microscopy, single-particle inductively coupled plasma mass spectroscopy, and total elemental analysis using ICP optical emission spectroscopy [159]. It is also essential to consider the potential impact of the detection method on the integrity of the plant samples and the interpretation of the results. Some considerations for selecting detection methods are listed in Table 6. These considerations involve both the characteristics of the particles being studied and the context in which the research is conducted. The choice between in vivo, in vitro, and in silico approaches is an important decision that influences the type of detection methods that can be applied.

### 3.1. Type of Detection Methods

Detecting the transport of nanoparticles in plants is an important area of research with several potential applications. There are various techniques and methods for detecting and studying nanoparticle transport in plants, as listed in Table 7. All of these methods play crucial roles in understanding how nanoparticles are taken up and distributed and how they interact with plant tissues. Zhang et al. reported the detection of 7 nm and 25 nm CeO_2_ NPs in the roots of Cucumis sativus by TEM [63]. They observed the adsorption and aggregation of the particles on the root surface, but not inside the cells [63]. Similarly, TEM was used to monitor the distribution of 25 nm CuO NPs in the roots of Glycine max, as shown in Figure 4 [167]. SEM could also be used to see the presence of particles in plant tissue. α-Fe_2_O_3_ NPs led to alterations in the root morphology of *Hordeum vulgare* and induced cell membrane injury and, hence, root damage, as indicated by SEM observations [168]. The use of several detection methods can mutually validate each other, thereby increasing the confidence level regarding particle transport. Bao et al. have shown that TEM analysis agreed well with the SP-ICP-MS results [169]. They also demonstrated that NPs accumulated in the leaf tissues showed large variations in size and relatively few readings compared to the NPs in the root tissues. Other examples of the application of detection methods for particle transport studies are shown in Table 8.

### 3.2. Challenges in Using Selected Detection Methods

Detecting nanoparticles in plants can be challenging due to the complex biological matrix and the small size of nanoparticles. Plants naturally contain a variety of elements and compounds that can interfere with the detection of nanoparticles, making it difficult to distinguish the nanoparticles from background signals [162,200]. The size of nanoparticles can be similar to or smaller than the cellular structures in plants [201]. This can make it challenging to visualize and accurately quantify nanoparticle uptake and distribution. Some nanoparticles may be present at low concentrations in plant tissues, requiring highly sensitive detection methods to accurately measure their presence [202]. Achieving high spatial resolution to precisely locate nanoparticles within plant tissues can be difficult, especially for in vivo imaging techniques. Some detection methods are destructive, requiring the disintegration of plant samples for analysis, which may limit the ability to conduct further studies on the same samples [203]. A summary of the advantages and disadvantages of each detection method can be seen in Table 9. The use of multiple detection methods, often referred to as a multi-method or multi-technique approach, can significantly enhance the robustness and reliability of research findings related to particle transport. Each detection method has its strengths and limitations, and combining several methods provides complementary information, cross-validation, and a more comprehensive understanding [184]. In a study conducted by Neves et al., the uptake and translocation of La_2_O_3_ NPs to the stems and leaves of *Pfaffia glomerata* (Spreng.) Pedersen were demonstrated after in vitro cultivation in the presence of 400 mg/L of NPs. Various detection methods, including laser ablation-ICP-MS (LA-ICP-MS) and μ-XRF, were employed. Both techniques proved to be effective for detecting nanoparticles in plants, yet LA-ICP-MS exhibited higher sensitivity than μ-XRF, enabling the improved detection and visualization of La distribution throughout the entire leaf [184].

## 4. Plant Stems as Future Recognition Sites for Transport

Introducing particles through the stem of a plant, as opposed to the roots or leaves, can offer several advantages. The vascular system in plant stems is better equipped to transport larger particles, such as nanoparticles, microspheres, or even macro-sized particles, compared to the relatively finer structures in roots or leaves. This means that when specific substances or materials need to be delivered in a particulate form, stem application allows for larger and more complex particles to be used. When larger particles are introduced through roots or leaves, there is a higher risk of blockage in the smaller vessels or stomata, potentially hampering nutrient and substance transport [219,220]. The stem’s larger vascular system can handle larger particles more efficiently, reducing the risk of clogs [221]. 

The vascular system in the stem, comprising the xylem and phloem, possesses specific characteristics that make it suitable for the introduction of larger particles compared to the roots or leaves [10,121]. The xylem and phloem vessels in the stem are larger in diameter compared to the corresponding structures in roots or leaves, as indicated in Table 10. This increased diameter allows for the accommodation of larger particles without clogging or obstructing the vascular pathways. The primary function of the vascular system is to transport water, nutrients, and other essential substances throughout the plant [29,222]. This natural transport system within the stem is well adapted to carry and distribute larger particles efficiently. The stem’s vascular system is responsible for the long-distance transport of water and nutrients from the roots to the leaves and other plant parts [223]. It is capable of maintaining the flow of materials over significant distances, making it an ideal route for distributing larger particles throughout the plant. The xylem and phloem vessels provide a continuous pathway throughout the stem, ensuring that introduced particles can be transported uniformly to different parts of the plant. The stem serves as a structural support for the plant, and this mechanical strength helps ensure that the vascular system remains intact even when larger particles are introduced, minimizing the risk of damage to the plant [224]. When particles are introduced through the roots or leaves, there is a greater risk of surface deposition, where the particles may remain on the outer surfaces and not penetrate the plant’s interior. In contrast, introducing particles through the stem offers a more direct and internal pathway for distribution. The vascular system in the stem typically transports materials in a unidirectional manner, following the natural flow of water and nutrients from the roots upward toward the leaves [225]. This consistent transport direction can be advantageous for certain applications.

These characteristics of the stem’s vascular system make it well suited for the introduction of larger particles, whether for the purpose of nutrient delivery, disease control, or other agricultural and horticultural practices. In our previous investigation, we successfully introduced 110 nm SiO_2_ nanoparticles into tomato seedlings using the stem cutting method [226]. However, it is essential to use appropriate techniques and precautions to ensure that the introduction of particles through the stem is carried out safely and effectively. 

**Table 10 nanomaterials-14-00131-t010:** Vascular bundle size in the stem, comprising the xylem and phloem, of various plant species.

Plants	Xylem Diameter (μm)	Phloem Diameter (μm)	Ref.
*Arabidopsis thaliana*	16	-	[227]
*Cucurbita maxima*	-	5	[228]
*Ipomoea hederifolia*	-	323–358	[229]
*Larix sibirica*	-	24–29	[230]
*Phaseolus vulgaris*	-	1.5–20	[228]
*Populus trichocarpa*	29–104	16–43	[231]
*Portulaca grandiflora*	121.5	-	[232]
*Portulaca oleracea*	98	-	[232]
*Portulaca quadrifida*	101.9	-	[232]
*Quercus chapmanni*	333	-	[233]
*Quercus falcata*	492	-	[233]
*Quercus hemisphaerica*	474	-	[233]
*Quercus incana*	470	-	[233]
*Quercus laevis*	469	-	[233]
*Quercus margaretta*	345	-	[233]
*Quercus myrtifolia*	455	-	[233]
*Quercus nigra*	517	-	[233]
*Quercus pubescens*	202.4	35.8	[234]
*Quercus sessiliflora*	1837.6	286.8	[235]
*Quercus austrina*	340	-	[233]
*Quercus geminata*	273	-	[233]
*Quercus michauxii*	332	-	[233]
*Quercus shumardii*	569	-	[233]
*Quercus virginiana*	286	-	[233]
*Ricinus communis*	300	100–150	[236]
*Rosmarinus officinalis*	467.3	7.2	[237]
*Solanum lycopersicum*	-	1.5–20	[228]
*Talinum fruticosum*	133.2	-	[232]
*Tectona grandis*	110.2–212.5	-	[238]
*Vitis vinifera*	33.6–35.8	-	[239]

## 5. Conclusions and Perspectives

The transport of nanoparticles into plants is indeed a growing area of research with a wide range of potential applications. Nanoparticles can be employed as delivery agents for a variety of substances, including nutrients, genetic material, or pesticides. Their small size, high surface area, and tunable properties make them useful for enhancing the targeted delivery of these materials to plants. However, it is important to consider factors such as nanoparticle toxicity, long-term effects, and environmental impact, as well to optimize the nanoparticles’ properties for specific delivery requirements. There are various methods for delivering nanoparticles to plants, and they can be broadly categorized into two main approaches: assisted delivery and passive delivery. Each method has its own set of advantages and disadvantages, depending on the specific application and goals of the research or practice. 

Passive delivery methods are more appealing due to their simplicity and reduced reliance on external sources, making them environmentally friendly and cost-effective. Roots and leaves are the primary sites for the passive delivery of exogenous materials into plants. These natural entry points have evolved to allow plants to absorb water, nutrients, and gases from their environment. However, both have natural barriers that can inhibit the delivery of nanoparticles or other exogenous materials to plants. These barriers can pose challenges in achieving efficient and reliable passive delivery. Exploring new introduction sites for delivering nanoparticles to plants is an interesting avenue of research, and the stem is indeed a potential candidate. Stems are natural conduits for the transport of water and nutrients within plants, as they contain vascular systems with xylem and phloem tissues. The large diameter of these vascular tissues suggests that they could potentially carry larger particles, making stems intriguing alternative introduction sites for nanoparticle delivery. The particle delivery process is also closely tied to the choice of detection methods, especially when it comes to studying nanoparticle localization and quantifying the number of particles in plant tissues. Utilizing multiple detection methods can enhance the reliability of the data obtained, as it allows for cross-validation and a more comprehensive understanding of the interaction between nanoparticles and plants. The plant effects produced by different nanomaterials vary greatly, and focus should be placed on the intrinsic mechanism of the effects of nanomaterials on plants, which can be studied in depth from the genetic and molecular levels to investigate the mechanism of the effects of nanomaterials on plant growth and development, as well as the plant’s response mechanism. The problem of considering stems as alternative introduction sites remains to be further investigated.

## Figures and Tables

**Figure 1 nanomaterials-14-00131-f001:**
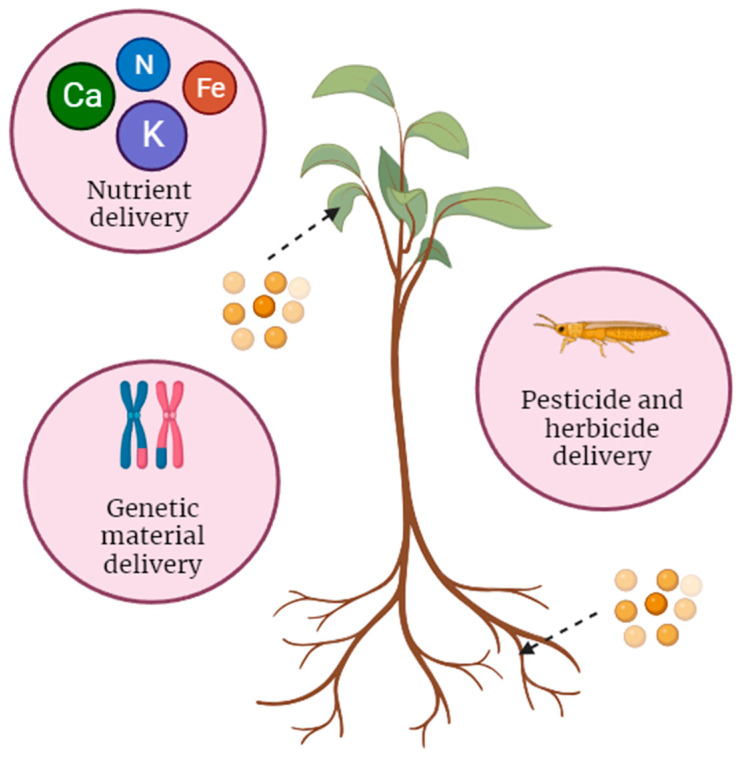
Nanoparticles used as carriers for nutrient (fertilizer), genetic material, and pesticide delivery into plants.

**Figure 2 nanomaterials-14-00131-f002:**
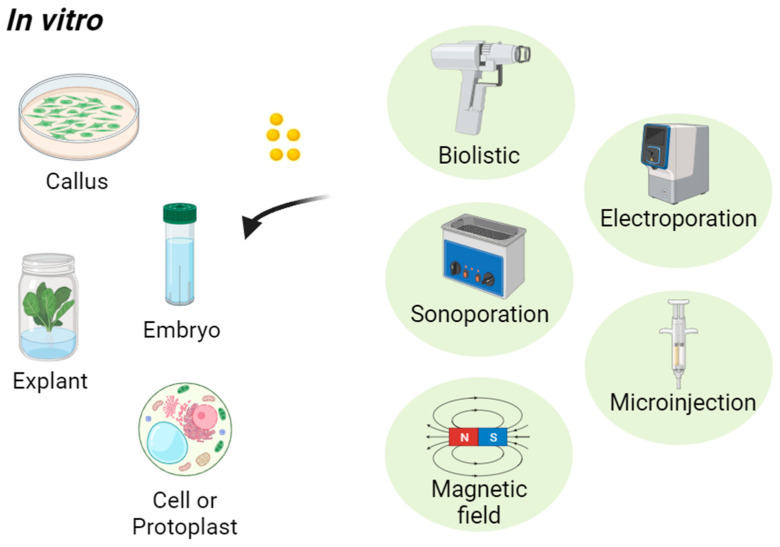
Assisted delivery techniques for introducing nanoparticles into plants in vitro.

**Figure 3 nanomaterials-14-00131-f003:**
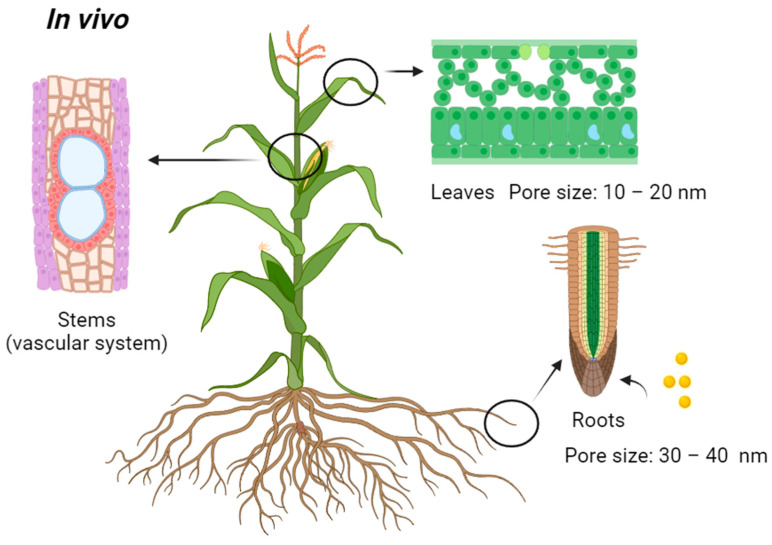
Passive delivery introduction sites for in vivo transport of nanoparticles into plants.

**Figure 4 nanomaterials-14-00131-f004:**
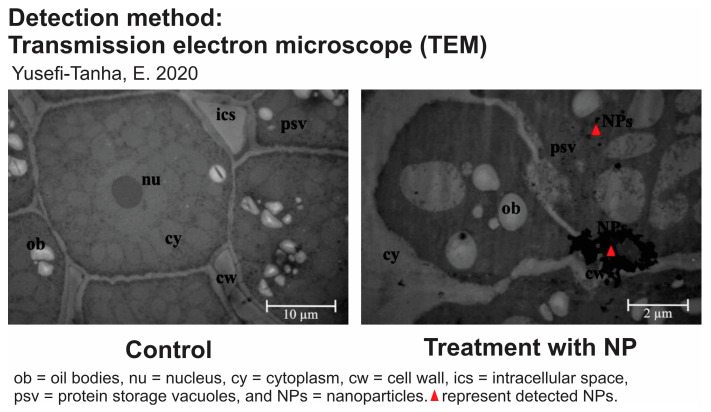
Transmission electron microscopy used to monitor the presence of NPs in plant tissue (adapted from [167]).

**Table 2 nanomaterials-14-00131-t002:** Types of nanoparticles used for transport in plants and their places of introduction.

Type	NPs	Size (nm)	Plants	Introduction Site	Ref.
Metal or metalloid	Ag	1–10	Tomato	Roots	[47]
	Ag	10	*Triticum aestivum*	Roots	[48]
	Ag	20 ± 3	*Linum usitatissimum*, *Lolium perenne*, *Hordeum vulgare*	Roots	[49]
	Ag	20	*Arabidopsis thaliana*	Roots	[50]
	Ag	10	*Phaseolus radiatus*, *Sorghum bicolor*	Roots	[51]
	Ag	10–15	*Lycopersicum esculentum*	Roots	[52]
	Ag	27.3 ± 6	*Populus deltoides*, *Arabidopsis thaliana*	Roots	[53]
	Al	18	*Brassica napus*, *Raphanus sativus*, *Lolium perenne*, *Lactuca sativa*, *Zea mays*, *Cucumis sativus*	Roots	[54]
	Au	1–3	*Oryza sativa*	Roots	[55]
	Au	3.5 and 18	*Nicotiana xanthi*	Roots	[56]
	Au	6–10	*Oryza sativa*, *Lolium perenne*, *Raphanus sativus*, *Cucurbita mixta*	Roots	[57]
	Co	28	Tomato	Roots	[47]
	Ni	28	Tomato	Roots	[47]
	Si	14	*Arabidopsis thaliana*	Roots	[58]
	Zn	35	*Brassica napus*, *Raphanus sativus*, *Lolium perenne*, *Lactuca sativa*, *Zea mays*, *Cucumis sativus*	Roots	[54]
Metal or metalloid oxide	CeO_2_	25	*Holcus lanatus*, *Diplotaxis tenuifolia*	Roots	[59]
	CeO_2_	8	*Glycine max*	Roots	[60]
	CeO_2_	8 ± 1	*Oryza sativa*	Roots	[61]
	CeO_2_	20 ± 2	*Solanum lycopersicum*	Roots	[62]
	CeO_2_	6.6 ± 1; 25.2 ± 2	*Cucumis sativus*	Roots	[63]
	Fe_3_O_4_	20–30	Tomato	Roots	[47]
	Fe_3_O_4_	8	*Cucurbita maxima*	Roots	[64]
	SiO_2_	10–20	*Chelidonium majus*	Leaves	[65]
	SiO_2_	20	*Cucumis sativus*	Leaves	[66]
	TiO_2_	20	Tomato	Roots	[47]
	TiO_2_	20 ± 5	*Triticum aestivum*	Roots	[67]
	TiO_2_	27 ± 4	*Cucumis sativus*	Roots	[68]
	TiO_2_	27	*Lycopersicum esculentum*	Roots	[52]
	TiO_2_	2.8 ± 1	*Arabidopsis thaliana*	Roots	[69]
	ZnO	10	*Glycine max*	Roots	[60]
	ZnO	20 ± 5	*Brassica napus*, *Raphanus sativus*, *Lolium perenne*, *Lactuca sativa*, *Zea mays*, *Cucumis sativus*	Roots	[54]
Carbon-based	C	20	*Zea mays*	Roots	[70]
	Carbon nanotubes (CNT)	10–30	*Cicer arietinum*	Roots	[71]
	Multi-walled carbon nanotubes (MWCNT)	10–20	*Brassica napus*, *Raphanus sativus*, *Lolium perenne*, *Lactuca sativa*, *Zea mays*, *Cucumis sativus*	Roots	[54]
	MWCNT	6–9	*Zea mays*	Roots	[72]
	MWCNT	4–13	*Lactuca sativa*, *Oryza sativa*, *Cucumis sativus*, *Amaranthus tricolor*, *Abelmoschus esculentus*, *Capsicum annuum*, *Glycine max*	Roots	[73]
	MWCNT	30	*Brassica juncea*	Roots	[74]
	MWCNT	6–13	*Triticum aestivum*	Roots	[75]
	Single-walled carbon nanotubes (SWCNT)	20	*Nicotiana benthamiana*	Leaves	[76]
Semiconductor	3-mercaptopropionic acid (MPA) quantum dots (QDs)	4–5.4	*Lemna minor*	Leaves	[77]
	Cd-based QDs	1.9 and 2.4	*Allium cepa*	Roots	[78]
	CdSe/CdZnS QDs	19.5 ± 7	*Populus deltoides*	Roots	[79]
	CdTe QDs	4	*Oryza sativa*	Roots	
	Glutathione (GSH) QDs	4–4.4	*Lemna minor*	Leaves	[77]
Polymeric	Chitosan	19–21	*Oryza sativa*	Roots	[80]
	Thiamine loaded chitosan	10	*Cicer arietinum*	Roots	[81]
Magnetic	Superparamagnetic iron oxide (SPION)	9	*Glycine max*	Roots	[82]

**Table 4 nanomaterials-14-00131-t004:** Advantages and disadvantages of using assisted delivery methods to introduce nanoparticles into plants.

Method of Assisted Delivery	Advantages	Disadvantages
Biolistic (gene gun)	- Can be applied to a wide range of plant species. - No need for specific plant developmental stages [124].	- Can cause physical damage to plant cells, potentially leading to cell death or reduced regeneration efficiency [125]. - Requires specialized equipment. - The composition of the delivery buffer can influence the stability of nanoparticles and their tendency to aggregate [126].
Sonoporation	- Non-invasive method, making it less damaging to plant cells. - Pores created are temporary and typically reseal over time [127].	- Optimization can be complex and may require extensive experimentation. - May have limited penetration into deep tissues [128]. - The composition of the medium in which sonoporation is conducted can influence nanoparticle stability [129].
Magnetic field	- Enable highly targeted and localized delivery to specific plant tissues or cells [97]. - Allows for precise control over the release of nanoparticles.	- Require the use of magnetic nanoparticles, which can limit their applicability [130]. - Methods are well suited for in vitro applications, and not in vivo.
Electroporation	- Can be used for a broad range of plant species. - Relatively fast, allowing for efficient delivery within a short time frame [131].	- Can lead to physical damage to the cell membranes, potentially causing cell death or reducing cell viability [132]. - Can induce cellular stress responses.
Microinjection	- Allows for precise and targeted delivery of nanoparticles into specific cells or locations within an organism. - Can be used to introduce a wide range of nanoparticles, making it a versatile technique [133].	- Can be invasive and potentially damaging to the plant cells, which may lead to stress or mortality. - Low-throughput method, making it unsuitable for large-scale applications [134].

**Table 5 nanomaterials-14-00131-t005:** Advantages and disadvantages of using existing passive delivery methods and plant developmental stages to introduce nanoparticles into plants.

Passive Delivery	Advantages	Disadvantages
Methodology		
Root uptake	- Plant roots are naturally adapted to take up water and nutrients from the soil. Nanoparticles can leverage these existing uptake mechanisms, making it a relatively non-invasive and eco-friendly method [29]. - Nanoparticles introduced through the root system can potentially provide longer-lasting effects compared to foliar applications, as they may be less prone to weathering and degradation [148].	- The transport of nanoparticles through the root system may be subject to size and charge limitations, as not all nanoparticles can efficiently pass through the root cell walls or move within the plant [142,145]. - The efficiency of nanoparticle uptake through roots can vary among different plant species. Some plants may not readily take up nanoparticles or may do so less efficiently [29,149].
Foliar uptake	- Leaves are the primary sites for photosynthesis and gas exchange in plants. Nanoparticles applied to leaves can be directly taken up by the plant, potentially leading to faster effects compared to root uptake [150]. - Leaf-based applications eliminate the need for nanoparticles to interact with soil components, which may affect the stability and bioavailability of the nanoparticles [151].	- Not all nanoparticles are efficiently taken up by leaves, and the extent of uptake can vary among plant species. Leaf properties, such as waxes and surface structures, can create barriers to nanoparticle penetration [26,149]. - Nanoparticles may not efficiently translocate to other plant tissues beyond the treated leaves, limiting their systemic effects [152].
Plant developmental stages		
Seed germination	- Nanoparticles become integrated into the plant from the earliest stages, potentially influencing overall development [33]. - Since seeds are typically uniform in size and structure, introducing nanoparticles during germination allows for relatively uniform exposure across a population of plants [153].	- During germination, the developing seedling may have limited root development, potentially limiting the uptake of nanoparticles [54]. - Nanoparticles introduced during germination might have a higher chance of causing toxicity, as developing seedlings are often more sensitive to external factors [154].
Seedling growth	- Seedlings generally have more developed root systems compared to germinating seeds, allowing for potentially higher uptake of nanoparticles [155]. - Seedlings are often more robust and resilient compared to germinating seeds, making them potentially better equipped to handle the introduction of nanoparticles without adverse effects on growth [155].	- Seedlings may exhibit variability in size and developmental stage, leading to potentially inconsistent exposure to nanoparticles across a population [156]. - Growing seedlings requires more resources such as space, light, and nutrients compared to germinating seeds [33].

**Table 6 nanomaterials-14-00131-t006:** Considerations for selecting detection methods for studying particle transport in plants.

Factors	Description
Methodological considerations	In vivo (within the living organism): Advantages: Provides real-time information about particle transport in a physiological context [160]. Considerations: May be challenging to observe and quantify; has potential effects on plant health [122]. In vitro (outside the living organism, typically in a controlled environment): Advantages: Allows for controlled experiments; easier to monitor and quantify [107]. Considerations: May not fully represent the complexity of the in vivo environment [108]. In silico (computer simulation): Advantages: Enables modeling and simulation of particle transport; cost-effective and flexible [161]. Considerations: Requires accurate input parameters; simplifications may limit realism [161].
Particle properties	The size and composition of nanoparticles may influence the choice of detection method. Some methods are better suited for specific sizes or materials [162].
Objectives	Localization: Detect nanoparticles on the plant’s surface, in the root, or within various plant tissues [97,163]. Quantification: Quantitative data on the concentration of nanoparticles in plant tissues or qualitative information on their presence [159,163].
Non-invasive vs. invasive	Some methods are non-invasive and allow real-time monitoring, while others require destructive sampling [164].
Sensitivity and precision	Considers the required sensitivity to detect low concentrations of nanoparticles [165].
Sample preparation	Evaluates the ease and compatibility of sample preparation with the chosen method. Some methods may require complex sample processing [166].

**Table 7 nanomaterials-14-00131-t007:** Techniques and methods used to detect nanoparticles in plants.

Methods	Description
Transmission electron microscopy (TEM)	TEM allows us to visualize nanoparticles at the nanoscale within plant tissues. By preparing ultrathin sections of plant material and using TEM, the internal distribution and movement of nanoparticles can be observed in different plant structures [167].
Scanning electron microscopy (SEM)	SEM is another microscopy technique that can be used to study the surface morphology of plant tissues and detect the presence of nanoparticles. It provides high-resolution images and can help identify the localization of nanoparticles on the plant’s surface [168].
Inductively coupled plasma mass spectrometry (ICP-MS)	ICP-MS is a highly sensitive analytical technique used to quantify the elemental composition of samples. By digesting plant tissues and then subjecting them to ICP-MS analysis, the presence and concentration of specific nanoparticles can later be determined [169].
Confocal laser scanning microscopy (CLSM)	CLSM is a non-destructive imaging technique that can be used to track the movement of fluorescently labeled nanoparticles within plant tissues over time. This method is particularly useful for studying the dynamics of nanoparticle transport [163].
X-ray fluorescence (XRF)	XRF can be employed to analyze the elemental composition of plant tissues and identify the presence of nanoparticles. It can provide information about the distribution of specific elements, including those contained in nanoparticles [33].
Fluorescence and luminescence techniques (Flo-Lum)	Utilizing fluorescent or luminescent tags on nanoparticles, researchers can track the movement of nanoparticles through plants using fluorescence microscopy or other imaging techniques [170].

**Table 8 nanomaterials-14-00131-t008:** Application of several detection methods for particle transport studies.

Detection Method	Detected NPs	Size (nm)	Plants	Introduction Site	Ref.
TEM	Ag	7 and 18	*Medicago sativa*	Roots	[171]
	Ag	10	*Arabidopsis thaliana*	Roots	[169]
	Au	3.5 and 18	*Nicotiana xanthi*	Roots	[56]
	CeO_2_	6.6 ± 1; 25.2 ± 2	*Cucumis sativus*	Roots	[63]
	MgO	27.7	Watermelon	Leaves	[172]
	SiO_2_	30	Cotton	Roots	[173]
	TiO_2_	24.5	Watermelon	Leaves	[172]
	TiO_2_	27 ± 4	*Cucumis sativus*	Roots	[68]
	ZnO	20	Ryegrass	Roots	[174]
	ZnO	30 ± 5	*Zea mays*	Roots	[175]
SEM	α-Fe_2_O_3_	14	*Hordeum vulgare*	Roots	[168]
	Ag@CoFe_2_O_4_	10	*Triticum aestivum*	Roots	[176]
	Ag_2_S	35	*Cucumis sativus, Triticum aestivum*	Roots	[177]
	CeO_2_	15.5	*Raphanus sativus*	Roots	[178]
	Fe_3_O_4_	12–20	*Lactuca sativa*	Roots	[179]
	MgO	15–20	*Arachis hypogaea*	Roots	[180]
	TiO_2_	19	*Oryza sativa*	Roots	[159]
	TiO_2_	12–20	*Lactuca sativa*	Roots	[179]
ICP-MS	Ag	10	*Arabidopsis thaliana*	Roots	[169]
	Ag	7.6 ± 2	*Oryza sativa*	Roots	[181]
	Au	2	*Oryza sativa*	Roots	[55]
	CeO_2_	30	*Solanum lycopersicum, Cucumis sativus, Cucurbita pepo, Glycine max*	Roots	[182]
	CeO_2_	30	*Raphanus sativus*	Roots	[183]
	CuO	25	*Raphanus sativus*	Roots	[183]
	La_2_O_3_	20–30	*Pfaffia glomerata*	Roots	[184]
	TiO_2_	30	*Raphanus sativus*	Roots	[185]
CLSM	CeO_2_	1.7–18	*Gossypium hirsutum, Zea mays*	Leaves	[186]
	MSNs	20	Lupin, wheat, maize	Roots	[163]
	QD	10	*Arabidopsis thaliana*	Leaves	[187]
	SiO_2_	1.7–18	*Gossypium hirsutum, Zea mays*	Leaves	[186]
	ZnO	30	*Triticum aestivum*	Leaves	[188]
XRF	Au	13.4 ± 1	*Arabidopsis thaliana*	Roots	[189]
	CeO_2_	12 ± 3	*Triticum aestivum*	Roots	[190]
	CeO_2_	4	*Zea mays, Lactuca sativa, Solanum lycopersicum, Oryza sativa*	Roots	[190]
	CuO	30.7 ± 4	*Oryza sativa*	Roots	[191]
	SiO_2_	10	*Solanum lycopersicum*	Roots	[33]
	TiO_2_	14 and 25	*Brassica napus, Triticum aestivum*	Roots	[192]
	TiO_2_	5–10	*Salvinia natans*	Roots	[193]
	TiO_2_	30	*Pisum sativum*	Roots	[194]
	TiO_2_	14	*Oryza sativa*	Roots	[195]
	ZnO	24.5 ± 4	*Oryza sativa*	Roots	[191]
Flo-Lum	Ag	35 ± 15	*Stevia rebaudiana*	Roots	[170]
	CdSe/ZnS QD	6.3 ± 1	*Arabidopsis thaliana*	Roots	[196]
	CeO_2_	8 ± 1	*Zea mays*	Roots	[197]
	TiO_2_	8	*Spirodela polyrrhiza*	Roots	[198]
	TiO_2_	2.8 ± 1	*Arabidopsis thaliana*	Roots	[199]

**Table 9 nanomaterials-14-00131-t009:** Advantages and disadvantages of detection methods for localizing or quantifying nanoparticles in plants.

Detection Method	Advantages	Disadvantages
TEM	- Provides incredibly high resolution. Can detect and characterize very small particles [204]. - Offers detailed structural information about the particles, including their size, shape, and arrangement [204].	- Sample preparation is complex and time-consuming. It often involves ultrathin sectioning or sample staining [205]. - Has a limited field of view, which means that only a small portion of the sample can be imaged at high resolution at a time [205].
SEM	- SEM has a larger field of view compared to TEM. Can accommodate a wide range of samples [206]. - Energy-dispersive X-ray spectroscopy (EDS) can be integrated with SEM for elemental analysis, helping to identify the composition of particles [207].	- Cannot reveal internal structures or details within particles unless the sample is specially prepared by cutting or fracturing [208]. - Non-conductive samples may require special preparation techniques, such as coating with a conductive layer, to avoid charging effects [208].
ICP-MS	- Extremely sensitive and can detect trace levels of elements and isotopes in a wide range of samples, making it suitable for both qualitative and quantitative analyses [209]. - Can analyze a wide range of elements, from the lightest (e.g., hydrogen) to the heaviest (e.g., uranium) elements in the periodic table [209].	- Sample preparation can be complex and time-consuming, particularly for solid samples like nanoparticles. Sample digestion and extraction methods may be required [210]. - Can be sensitive to interferences and matrix effects that may affect the accuracy of results, requiring the use of correction techniques [210].
CLSM	- Enables optical sectioning, allowing for the collection of images at various depths within a sample [211]. - Use of a pinhole aperture to restrict the collection of light from outside the focal plane, which results in reduced background noise and improved signal-to-noise ratio [212].	- Continuous laser illumination can lead to photobleaching of fluorescent dyes, reducing the fluorescence signal over time [213]. - Proper sample preparation is essential, including fixation, labeling, and mounting, which can be time-consuming [213].
XRF	- Non-destructive method; it allows for analysis without altering or damaging the samples [214]. - Can analyze broad range of elements simultaneously, providing comprehensive view of the elemental composition of the sample [214].	- Has limited spatial resolution, which can be a drawback when studying nanoparticles at the cellular or sub-cellular level [215]. - Provides information from the surface of the sample, and depth profiling may be limited [215].
Flo-Lum	- Allows the simultaneous detection of multiple particles or structures by using different fluorescent labels with distinct emission spectra [216]. - Enable real-time imaging and tracking of dynamic processes in live samples, making them suitable for cellular and molecular studies [217].	- Continuous exposure to excitation light can lead to photobleaching of fluorescent dyes, reducing signal intensity over time [218]. - When multiple fluorophores are used, bleed-through and crosstalk between channels can complicate data analysis [218].

## Data Availability

All data generated or analyzed during this work are included in this published article.

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
