# Peer review of "Transport of Nanoparticles into Plants and Their Detection Methods"

_nanomaterials, 2024, doi:10.3390/nano14020131_

Round 1

Reviewer 1 Report

Comments and Suggestions for Authors

The manuscript titled “Transport of Nanoparticles in Plants and Their Detection Methods” by Sembada A.A., & Lenggoro, I.W. is a Review work where the authors show the most recent advances in the use of nanoparticles to deliver compounds in plants and their associated transport mechanisms inner the plant tissues and the characterization techniques. The paper is interesting and could gain impact in the near future.

However, it exists some points that need to be addressed (please, see them below detailed point-by-point). The most relevant outcomes remarked by the authors can contribute in the growth of many fields like the design of the next-generation of agricultural crops enhancing the plant productivity. For this reason, I will recommend the present scientific manuscript for further publication in Nanomaterials once all the below described suggestions will be properly fixed.

Here, there exists some points that must be covered in order to improve the scientific quality of the manuscript paper:

1) KEYWORDS. (OPTIONAL) The authors should consider to add the terms “nanoparticle characterization” and “detection strategies” in the keyword list.

2) INTRODUCTION. “The primary processes involved (…) water uptake and nutrient uuptake” (lines 30-31). Please, the authors should rephrase the end of this statement in order to avoid redundancies. Similar comment for the following sentence “The root hairs (…) play a critical role in water uptake (…) root haris play a significant role (…)” (lines 34-35). “In this review, we will summarize (…) We also summarize the challenges (…)” (lines 91-93) is other example where the authors should also revise. This point should be taken into account for the rest of the main manuscript body text.

3) “The transport of nanoparticles into plants (…) depending on the type of nanoparticles and the intended purpose” (lines 67-71). Do the authors refer to the nanoparticle nature (chemistry) or the nanoparticle size dimensions? A brief statement should be provided in order to furnish further insights.

4) TRANSPORT NANOPARTICLES IN PLANTS. “Nanoparticles can be designed to transport specific materials, compounds, or genetic information within plants (…) Tombuloglu et al. succeeded in sinthesizing micro-nutrient nanoparticles (NPs) composites applied to (…)” (lines 95-102). What is the expected release yields in plants of the use of these NPs? Some quantitative information should be provided in this regard.

5) Table 1 (line 152). I agreee with the considerations made by the authors about the information concerning to this Table. Nevertheless, it should not be neglected the effect of the soil pH, temperature and relative humidity as additional “Environmental considerations”.

6) Table 2 (line 159). Size: 20 ± 2.5 nm (Ag NPs) or Size: 20 ± 1.9 nm (CeO2 NPs), among many other examples. Please, the authors should homogenize the significant figures. Similar comment for the Table 3 (line 227) è Size: 123 ± 21.3 nm (hPAMAM-G2 NPs).

7)  Then, the authors should also briefly discuss about the potential impact of the use of certain NPs in the plant development and growth [1].

[1] Faizan, M.; et al. Nanobionics: A Sustainable Agricultural Approach towards Understanding Plant Response to Heavy Metals, Drought, and Salt Stress. Nanomaterials 2023, 13, 974. https://doi.org/10.3390/nano13060974.

8) “Magnetic nanoparticles can be guided (…) using external magnetic fields (…)” (lines 202-203). Here, this Review work should also introduce bulk [2] and single molecule [3] techniques to characterize the intrinsic magnetic properties of the NPs.

[2] Calvo, R.; et al. Novel Characterization Techniques for Multifunctional Plasmonic-Magnetic Nanoparticles in Biomedical Applications. Nanomaterials 2023, 13, 2929. https://doi.org/10.3390/nano13222929.

[3] Winkler, R.; et al. A Review of the Current State of Magnetic Force Microscopy to Unravel the Magnetic Properties of Nanomaterials Applied in Biological Systems and Future Direction for Quantum Technologies. Nanomaterials 2023, 13, 2585. https://doi.org/10.3390/nano13182585.

9) “Most of the assisted delivery method are used for transporting nanoparticles in vitro (…) laboratory setting” (lines 207-210). Please, the authors should briefly argue about the scalability of these in vitro strategies at larger scales.

10) Figure 3 (line 247). (OPTIONAL) The authors should consider to add the mean pore sizes that can be found in the plant leaves, stems and roots.

11) Table 4 (line 265). The authors outlined some weaknesses linked to the existing assisted delivery methods. The potential nanoparticle aggregation under certain conditions should be considered? (this points was already briefly discussed by the authors in the lines 320-321).

12) “Passive delivery methods (…) such as mechanical or electrical forces (…)” (lines 269-271). Similar than above described, the authors should provide techniques to fully characterize the nanoparticle mechanical properties in bulk [4] and at the nanoscale [5] and also their electrical response [6] which is crucial to better understand their performance once the NPs have been uptaked by the plant tissues.

[4] Narita, F.; et al. Multi-Scale Analysis and Testing of Tensile Behavior in Polymers with Randomly Oriented and Agglomerated Cellulose Nanofibers. Nanomaterials 2020, 10, 700. https://doi.org/10.3390/nano10040700.

[5] Magazzù, A.; et al. Investigation of Soft Matter Nanomechanics by Atomic Force Micrsocopy and Optical Tweezers: A Comprehensive Review. Nanomaterials 2023, 13, 693. https://doi.org/10.3390/nano13060963.

[6] Okpara, E.C.; et al. Electrochemical Characterization and Detection of Lead in Water Using SPCE Modified with BiONPs/PANI. Nanomaterials 2021, 11, 1294. https://doi.org/10.3390/nano11051294.

13) DETECTION OF NANOPARTICLES IN PLANTS. Table 6 (line 312). The authors should also consider to add as relevant factors to be consider the possibility to conduct the research studies by “in vivo”, “in vitro” or “in silico”, respectively.

14) “Detecting nanoparticles in plants can be challenging due to the complex biological matrix and the small size of nanoparticles (…) from background signals” (lines 334-337). I agree with these information details furnished by the authors. For this reason, it may be opportune to present the possibility to exploit correlative detection techniques (e.g. SEM measurements coupled with fluorescence microscopy) to overcome some of the limitations depicted in Table 9.

15) PLANTS STEMS AS FUTURE RECOGNITION SITES FOR TRANSPORT. This section is clearly explained. No actions are requested from the authors.

16) CONCLUSIONS AND PERSPECTIVES. This section perfectly highlights the most relevant outcomes found in this Review work. The authors should remark some promising Industrial applications (which some of them were already indicated in the Introduction section) derived from the use of NPs in plant crops.

17) REFERENCES. The references are in the proper format style of Nanomaterials. No actions are requested from the authors.

Comments on the Quality of English Language

The manuscript is generally well-written. However, the authors need to check to fix final details susceptible to be improved.

Reviewer 2 Report

Comments and Suggestions for Authors

Title: Transport of Nanoparticles in Plants and Their Detection Methods

This manuscript provides an overview of the different transport pathways through which nanoparticles enter plants, including crops, fruits and vegetables, trees and shrubs, and more. It also discusses the challenges of existing detection methods and explores potential alternative approaches. Before accepting this review, the following points should be considered: (1) The title “The transport of nanoparticles in plants” and the presentation of related “The transport of nanoparticles into plants” content should be reviewed to avoid confusion. (2) It is important to highlight the novelty of this review in terms of its coverage of nanoparticle pathways into plants and their subsequent detection, distinguishing it from similar articles. (3) The interactions between plants and nanoparticles should be given due attention and emphasized throughout the manuscript.

Major:

Abstract & Keywords:

Lack of a systematic summary of the current status of research on the interaction of nanomaterials with plant tissues and cells, and the mechanisms of transport, translocation, and uptake of nanomaterials in plants.

L15 What are the “innovative approaches” expressed? Suggested additions.

L25 Adjust “xylem/phloem” to a more relevant keyword and suggest the inclusion of “nanoparticles”.

Introduction & literature review:

Please concisely write the background, highlighting important points.

The description of water, nutrients, and gas transport in plants is lengthy. In addition, is the description of basic concepts such as nanoparticles exhaustive?

L65-67 Suggested rewording to make it easier for readers to understand.

L94 “2. Transport of Nanoparticles in Plants” Please change the header to match the content or add background information to the content of the header.

L160 Research on the interaction of nanomaterials with plants has mainly focused on phytotoxicology. At the same time, less research has been conducted on positive effects such as increasing crop productivity and enhancing plant resistance, and research on beneficial effects on plants is still incomplete.

L184-186 Please add that nanoparticle transport in plants can be categorized based on two main mechanisms, i.e., assisted delivery and passive delivery.

L233 Additional clarification was suggested that assisted delivery and passive delivery are two different and separate concepts.

L241-243 in vivo” is not explained at first appearance.

Table 5 “Passive delivery methods” considers the addition of treatment stages and methods for passively introducing nanoparticles into plant tissues, such as plant seed germination or seedling growth stages.

L281-283 Rephrase the sentence to make it clearer.

Conclusions and Perspectives:

The plant effects produced by different nanomaterials vary greatly, and the focus should be on the intrinsic mechanism of the effects of nanomaterials on plants, which can be studied in depth from the genetic and molecular levels to investigate the mechanism of the effects of nanomaterials on plant growth and development, as well as the plant's response mechanism. The problem of considering stems as alternative introduction sites remains to be further investigated.

Minor:

L312 Table 6. spelled incorrectly. “Considerations for selecting detection methods for studying particle transport in plan” was changed to “… in plant”.

L299 “3. Detection of Nanoparticles in Plants” corresponds to the title number of L350 “3. Plant Stems as Future Recognition Sites for Transport”.

Please proofread the renumbering of the “Conclusions and Perspectives” section.

Reviewer 3 Report

Comments and Suggestions for Authors

Transport of nanoparticles in plants is an interesting topic. This work reviewed the transport of engineered nanomaterials in plants and their detection methods. This manuscript is well organized and the results were important to understand the fate and effects of nanoparticles. I think this work can be accepted after minor revisions.

1. The negative effects of engineered nanomaterials to plants should be discussed in a short paragraph.

2. The origin of nanoparticles should be divided into commercial and biogenic nanoparticles.

3. Some SEM/TEM images should be presented in addition to the virtual images.

Round 2

Reviewer 1 Report

Comments and Suggestions for Authors

The authors did a significant deal of effort to cover all the comments raised by the Reviewers. For this reason, the scientific manuscript quality was greatly improved. Based on the significance of the topic addressed by this Review work and the scope of Nanomaterials I warmly endorse this work for further publication in this journal. 

Reviewer 2 Report

Comments and Suggestions for Authors

This version can be accepted.